# Horizontal Temperature Fluxes in the Arctic in CMIP5 Model Results Analyzed with Self-Organizing Maps

**Daniel Mewes \***  **and Christoph Jacobi**

Leipzig Institute for Meteorology, Universität Leipzig, Stephanstr. 3, 04103 Leipzig, Germany;
jacobi@rz.uni-leipzig.de
\* Correspondence: daniel.mewes@uni-leipzig.de

**Abstract:** The meridional temperature gradient between mid and high latitudes decreases by Arctic amplification. Following this decrease, the circulation in the mid latitudes may change and, therefore, the meridional flux of heat and moisture increases. This might increase the Arctic temperatures even further. A proxy for the vertically integrated atmospheric horizontal energy flux was analyzed using the self-organizing-map (SOM) method. Climate Model Intercomparison Project Phase 5 (CMIP5) model data of the historical and Representative Concentration Pathway 8.5 (RCP8.5) experiments were analyzed to extract horizontal flux patterns. These patterns were analyzed for changes between and within the respective experiments. It was found that the general horizontal flux patterns are reproduced by all models and in all experiments in comparison with reanalyses. By comparing the reanalysis time frame with the respective historical experiments, we found that the general occurrence frequencies of the patterns differ substantially. The results show that the general structure of the flux patterns is not changed when comparing the historical and RCP8.5 results. However, the amplitudes of the fluxes are decreasing. It is suggested that the amplitudes are smaller in the RCP8.5 results compared to the historical results, following a greater meandering of the jet stream, which yields smaller flux amplitudes of the cluster mean.

**Keywords:** self-organizing maps; CMIP5; horizontal heat flux

## 1. Introduction

Arctic amplification describes enhanced warming of the Arctic relative to the rest of the world due to climate change [1–3]. Arctic amplification is characterized by the increase of atmospheric temperatures, as well as the decrease in sea-ice extent in the Arctic region. Due to these changes, during winter, the sea-level pressure decreases over the Arctic [4,5], which influences horizontal transports and fluxes. Of course, as Barnes and Polvani [6] have shown, Arctic amplification is not the sole reason for the change of circulation. However, due to the direct influence of Arctic warming on circulation changes and vice versa, investigation of horizontal energy fluxes is necessary for an understanding of climate change at high latitudes.

Analyzing the results from the Climate Model Intercomparison Project Phase 5 (CMIP5), Taylor et al. [7] have already shown that the low-latitude circulation will change in a warming climate. Feldl and Bordoni [8] found that the strength of the Hadley cell is reduced by up to 2.6% per 1 K temperature increase in case of an abrupt quadrupling of the carbon dioxide concentration. The change of the Hadley cell strength might also influence the circulation at mid and high latitudes [9,10]. This kind of response of large-scale atmospheric structures to a warmer climate was shown in the CMIP5 results of Vallis et al. [11]. For the Representative Concentration Pathway 8.5 (RCP8.5, [12]) scenario, they found that the surface westerlies are shifted poleward. This shift is connected to a Hadley cell expansion. In addition, they found that the westerlies are strengthened in winter.

Geoffroy et al. [13] showed that the horizontal heat transport is the main contributor to land–sea warming differences. On a global scale, atmospheric circulation reduces the mass transport by about 5% per 1 K temperature increase [14]. Focusing on the high latitudes, Kjellsson [14] showed that the total poleward energy transport increases, while the poleward mass transport decreases.

Horizontal heat flux is organized in different patterns with preferred flux paths. Analyzing and quantifying heat transport and its trends and changes therefore requires circulation pattern analysis. One of the recently preferred methods for this purpose is the self-organizing maps (SOM) method, which has already been used for clustering and for the extraction of circulation patterns at different spatial scales [15–21]. SOM is a simple unsupervised neural network that groups multidimensional data into a two-dimensional map of patterns without a priori assumptions [22].

Mewes and Jacobi [20] analyzed the horizontal moist static energy flux into the Arctic based on 39 years of ERA Interim [23] reanalyses. They used the SOM method to find distinct horizontal flux patterns that govern the exchange of energy between mid latitudes and the Arctic. Three intrinsic flux patterns were found, namely the Atlantic pathway, which is connected with fluxes of energy through the North Atlantic into the Arctic, the Siberian pathway, which is connected with fluxes through Siberia and the North American continent into the Arctic, and the Pacific pathway, which is connected with fluxes that originate from the North Pacific. According to their trend analysis, the Atlantic pathway occurrence frequency increased during the last 30 years, while the occurrence frequency of the Pacific pathway decreased. This, however raises the question of the extent to which these patterns are also represented in other data sets.Therefore, in order to investigate whether the flux patterns presented in Mewes and Jacobi [20] are consistent across different global climate models results and to analyze possible future trends in pattern occurrence, in this study, we cluster the horizontal energy flux of selected CMIP5 model results for two different scenarios—namely, an analysis of historical runs and future horizontal fluxes based on the RCP8.5 projection. The clustered patterns will be compared with, among others, the results of Mewes and Jacobi [20], i.e., with the classification based on ERA Interim reanalyses. The purpose of this work is to investigate if and how the CMIP5 data SOM patterns are relatable to the SOM pattern derived from ERA Interim reanalyses. Actually, as will be shown below, as in Mewes and Jacobi [20], three pathways can be identified in each of the CMIP5 models used in our study. Differences between the CMIP5 models are expected, e.g., with regard to the occurrence frequencies of pathways and their trends.

The remainder of this paper is organized as follows: In Section 2 the CMIP5 models will be briefly presented, and the methods applied for analysis, especially the SOM method, will be described. The results are presented in Section 3 and discussed in Section 4.

## 2. Method and Data

### 2.1. CMIP5 Model Data

Eight different models that participated in CMIP5 are used in this study. They are listed in Table 1 together with their respective horizontal resolutions, also expressed in kilometers at the equator.

**Table 1.** List of the Climate Model Intercomparison Project Phase 5 (CMIP5) models used, their respective resolutions (third column), and their resolutions in km at the equator (fourth and fifth columns).

| CMIP5 Model ID | Institution | | Horizontal Res. (°) | Latitude Res. (km) | Longitude Res. (km) |
|---|---|---|---|---|---|
| MRI-CGCM3 | MRI, Japan | Yukimoto et al. [24] | 1.1×1.1 | 120 | 120 |
| GFDL-CM3 | NOAA GDFL, USA | Collins et al. [25] | 2.5×2.0 | 275 | 220 |
| CMCC-CESM | CMCC, Italy | Vichi et al. [26] | 3.7×3.7 | 410 | 410 |
| CMCC-CMS | CMCC, Italy | Davini et al. [27] | 1.9×1.9 | 210 | 210 |
| MIROC-ESM | JAMSTEC, Japan | Watanabe et al. [28] | 2.8×2.8 | 310 | 310 |
| HadGEM2-CC | MOHC, UK | Collins et al. [29] | 1.9×1.2 | 210 | 130 |
| MPI-ESM-LR | MPI-M, Germany | Stevens et al. [30] | 1.9×1.9 | 210 | 210 |
| MPI-ESM-MR | MPI-M, Germany | Stevens et al. [30] | 1.9×1.9 | 210 | 210 |

All of these models are considered high-top models, which indicates a model top level above 1 hPa. We opted for these models, as the high-top models perform better at representing coupled stratosphere–troposphere processes, such as sudden stratospheric warmings [31]. As we are looking at daily data, this improved interaction between the lower and the middle atmosphere will improve the variability represented by the models. This important impact of the stratosphere on the troposphere has been shown in multiple studies (e.g., [32,33]).

For our study, we define two experimental baselines: (i) Historical, which is based on the CMIP5 historical runs, and (ii) RCP8.5, which is based on the CMIP5 Representative Concentration Pathway 8.5 runs. We analyze the boreal winters (December through February) from 1950 to 1999 for our historical studies, while the winters 2050–2099 are used for our RCP8.5 study.

The horizontal temperature flux $F = v\,T$ was analyzed at 500 hPa, where $v$ and $T$ are the horizontal wind vector and the temperature at 500 hPa, respectively. The horizontal temperature flux is taken here as a proxy for tropospheric energy fluxes into the Arctic, and has been found in previous analyses to be comparable to the vertically integrated (surface to 200 hPa) moist static energy flux that was analyzed in Mewes and Jacobi [20]. Furthermore, the analysis by Sorokina and Esau [34] of the Integrated Global Radiosonde Archive shows that the meridional energy transport across 70 degrees North into higher latitudes, in which we are interested, is strongest at 500 hPa.

$F$ was calculated based on daily winter data for each model and separately for the historical and RCP8.5 data. Furthermore, the vector field $F$ is confined to data north of $50° N$ to minimize calculation time and to keep the training of the SOM focused on the high latitudes of the Northern Hemisphere.

Additionally, we calculated a multi-model mean. This multi-model mean shall give the best estimate based on the group of models used [35].

## 2.2. Self-Organizing Maps

Intrinsic circulation patterns will be extracted from the temperature fluxes using the SOM method [22]. SOM is a simple neural network that groups multidimensional data into a two-dimensional map of patterns in an unsupervised manner and without a priori assumptions. This unsupervised learning approach is an advantage over commonly used approaches like Principle Component and (rotated) Empirical Orthogonal Function analysis, as it is not preconditioned through linear assumptions [18,36]. Like most cluster analysis tools, SOM maximizes the differences between clusters while minimizing the differences within clusters. However, due to the special characteristic of the SOM that forces each cluster to also learn with dependence on its neighbors, the clusters (describing the circulation patterns) evolve in such a way that patterns that are closer together (farther apart) in the map are also more similar (less similar) to each other. The SOMs were calculated with the python package "somoclu" [37].

The SOM method was chosen over other clustering methods (e.g., k-means) because of the map feature. This ordering of patterns in such a way that similar (dissimilar) patterns are closer (farther apart) within the map helps to cover the continuous changes of the atmosphere and thus resembles the non-discrete nature of the atmosphere, in contrast to other methods.

The choice of the number of clusters is a crucial step in our analysis. In a previous study, Mewes and Jacobi [20] used an SOM size of four columns and three rows, which presents enough variability without losing too many specific features. We chose the four-by-three SOM size according to the experiments described in an internal report by Mewes and Jacobi [38]. As described in Mewes and Jacobi [20], the twelve clusters of each SOM will be manually combined in specific flux pathways only. The common criteria are the general amplitudes, direction, and rotation of the vector field of $F$. This will help to group common temperature flux features even further so that we end up with few flux pathways. The manual grouping of the patterns is a method that is common in the literature (e.g., [39,40]), and is applied to facilitate the interpretation and discussion. This manual grouping is necessary due to the underlying Euclidean distance of the vectorized fields, with which the SOM decides the classification of one day of data to a distinct pattern. The Euclidean distance method

might categorize data into a certain transport pathway, while from a meteorological point of view (amplitudes, direction, and rotation of the vector field), it would not fit into that pathway. Without this manual step, patterns may be assigned to a group with which they would not share any features. With the chosen approach of manual grouping, we can be more certain that patterns that actually share common features (such as amplitudes, direction, and rotation of $F$) are within the group to which they belong with respect to these mentioned criteria. Eventually, the Atlantic, the Continental, and the Pacific pathways, as introduced by Mewes and Jacobi [20], are calculated for each model by weighting the respective patterns with their relative occurrence frequencies. Our analysis shows that these patterns are the dominant patterns in all models.

The emerging patterns of each model's SOM are calculated and manually and separately grouped for the historical and RCP8.5 runs. This results in a set of sixteen-times-three pathways based on eight-times-two SOMs, with four columns and three rows each.

To investigate how pathways might differ between historical and RCP8.5 runs, we also mapped the RCP8.5 data onto the historical SOMs. This was done by taking the respective SOMs derived from the historical data and projecting the RCP8.5 data onto the historical SOM patterns. Thereby, the learning of the network is deactivated, which results in the RCP8.5 data being mapped onto the historical SOM only. This is equivalent to an assumption that the general SOMs do not change between the historical and RCP8.5 time frames. The general structure and the changes in relative occurrence frequencies of the pathways are compared, as will be shown below.

To further validate the different mappings of the RCP8.5 data based on the historical and the RCP8.5 SOMs, we use the V-measure [41]. Mapping means here that, for each daily data field of the RCP8.5 data, we are looking for the SOM pattern that has the smallest Euclidean distance to the data on this day. After the mapping onto the respective SOM, each SOM pattern consists of a set of pattern members. Eventually, each daily data field is part of a pattern within an SOM, and thus becomes a member of this pattern. The V-measure [41] describes the qualitative agreement of the two independent mappings of the RCP8.5 data onto the historical SOM and the RCP8.5 SOM. It ranges from 0 to 1. If the V-measure is equal to 1, the mappings are identical. The V-measure does not compare the resulting patterns themselves; rather, it compares the mapping, i.e., the correspondence of one daily datum to a pattern. The V-measure is derived from the homogeneity score and the completeness score [41]. These scores show a defined view of how the two mappings differ in the two defined values of homogeneity and completeness. A homogeneity of 1 is reached if one pattern of a mapping only contains members from one single pattern of a reference mapping. A completeness of 1 is achieved if all members of one single pattern of a reference mapping are assigned to one pattern of another mapping.

## 3. Results

### 3.1. Historical Patterns

Each of the patterns were grouped into the three main pathways named: The Atlantic pathway, the Continental pathway, and the Pacific pathway. Figures 1–3 show the three pathways based on the historical data for each analyzed CMIP5 model. The individual patterns emerging from each SOM can be seen in the Supplementary Figures S1–S8. The Atlantic pathway mainly features fluxes originating from the North Atlantic reaching into the Arctic. The Continental pathway is described by weak meridional transports and a more zonal flow. Transports that originate from the North Pacific into the Arctic are grouped into the Pacific pathway. Some patterns that emerged from the SOM cannot be easily sorted into either the Atlantic, the Continental, or the Pacific pathway. These patterns that do not fit this three-pathway definition will be omitted in the discussion, as they resemble superpositions of the other pathways. Subsequently, they will be named as miscellaneous pathways. Generally, each model's SOM contains at least one of these miscellaneous pathways. The introduction of the miscellaneous pathways was necessary, as the superpositions shown by the models were not uniform across the models. This made it non-trivial to group the superpositions into one of the three

major pathways without misrepresenting the key features that the Atlantic, Continental, and Pacific pathways describe.

As an example of how this manual grouping was performed, we will look at the pattern 1.1 of Supplementary Figure S3. This description shall briefly describe our approach on how these patterns are grouped into the categories. This pattern was not put into the Pacific pathway category, as the general flux from the North Pacific is relatively low compared to patterns that are considered Pacific pathways (compare patterns 2.1 and 3.1 of Figure S3). However, the strongest flux into higher latitudes occurs over Greenland, but its general direction does not reach the central Arctic region, as it turns southward again at the east coast of Greenland. Pattern 1.1 of Figure S3 does not show specifically strong or consistent zonal fluxes either, which would put it into the Continental pathway category. Thus, pattern 1.1 of Figure S3 is considered a miscellaneous pattern.

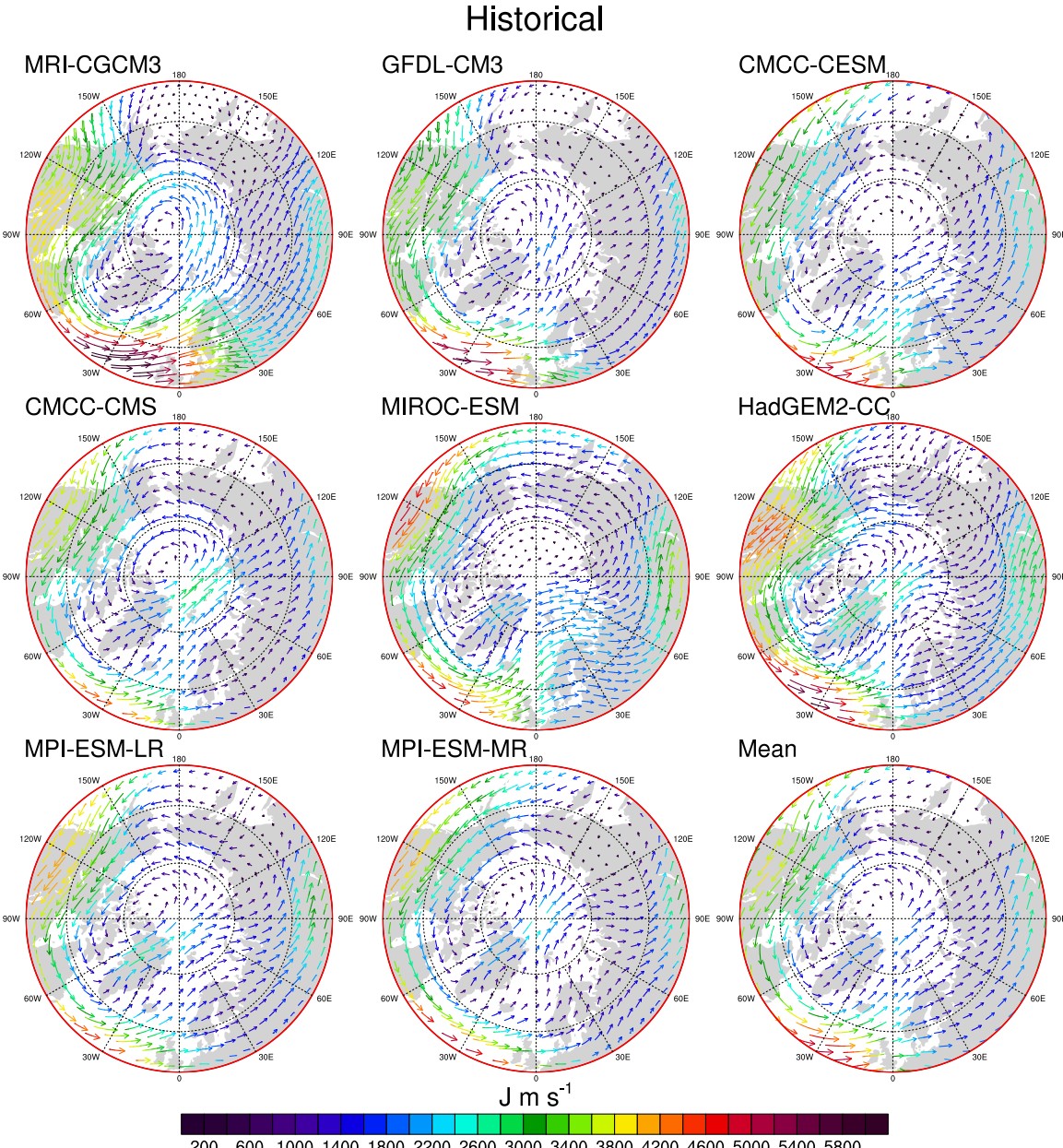

**Figure 1.** The Atlantic pathway for each of the models listed in Table 1, derived based on the CMIP5 historical data for the winters from 1950–1999. Colors and arrows show the amplitudes and directions of the horizontal temperature flux at 500 hPa. The multi-model mean of all pathways is shown in the bottom-right map.

The structures of the respective pathways show differences between the models. The Atlantic pathway usually shows fluxes mainly over Ny-Ålesund into the Barents Sea and Kara Sea regions (e.g., Figure 1: MRI-CGCM3), or fluxes mainly over Greenland and the Fram Strait into the Laptev Sea (e.g., Figure 1: MPI-ESM-MR). In general, the Atlantic pathway representation for all models is connected with fluxes from the North Atlantic into the central Arctic region, and has been shown to be connected with a warm Arctic and cold continent pattern [20,42]. The so-called Continental pathway is mostly connected with weak meridional transports (e.g., Figure 2: CMCC-CMS, GFDL-CM3), with some exceptions when meridional transports mainly originate from the North of the Eurasian continent or from North America (e.g., Figure 2: MRI-CGCM3, MIROC-ESM). It is connected with negative temperature anomalies over the central Arctic and positive temperature anomalies over North America and Siberia. The Pacific pathway originates from the North Pacific ocean and enters the Arctic through the Bering Strait and East Siberia (e.g., Figure 3: GFDL-CM3, CMCC-CMS). In some cases, the transport through the Bering Strait is negligible compared to the transport over East Siberia (e.g., Figure 3: HadGEM2-CC, MPI-ESM-LR). For a direct comparison with reanalysis results, Figure 4 shows the pathways of the ERA-Interim reanalysis data analyzed by Mewes and Jacobi [20].

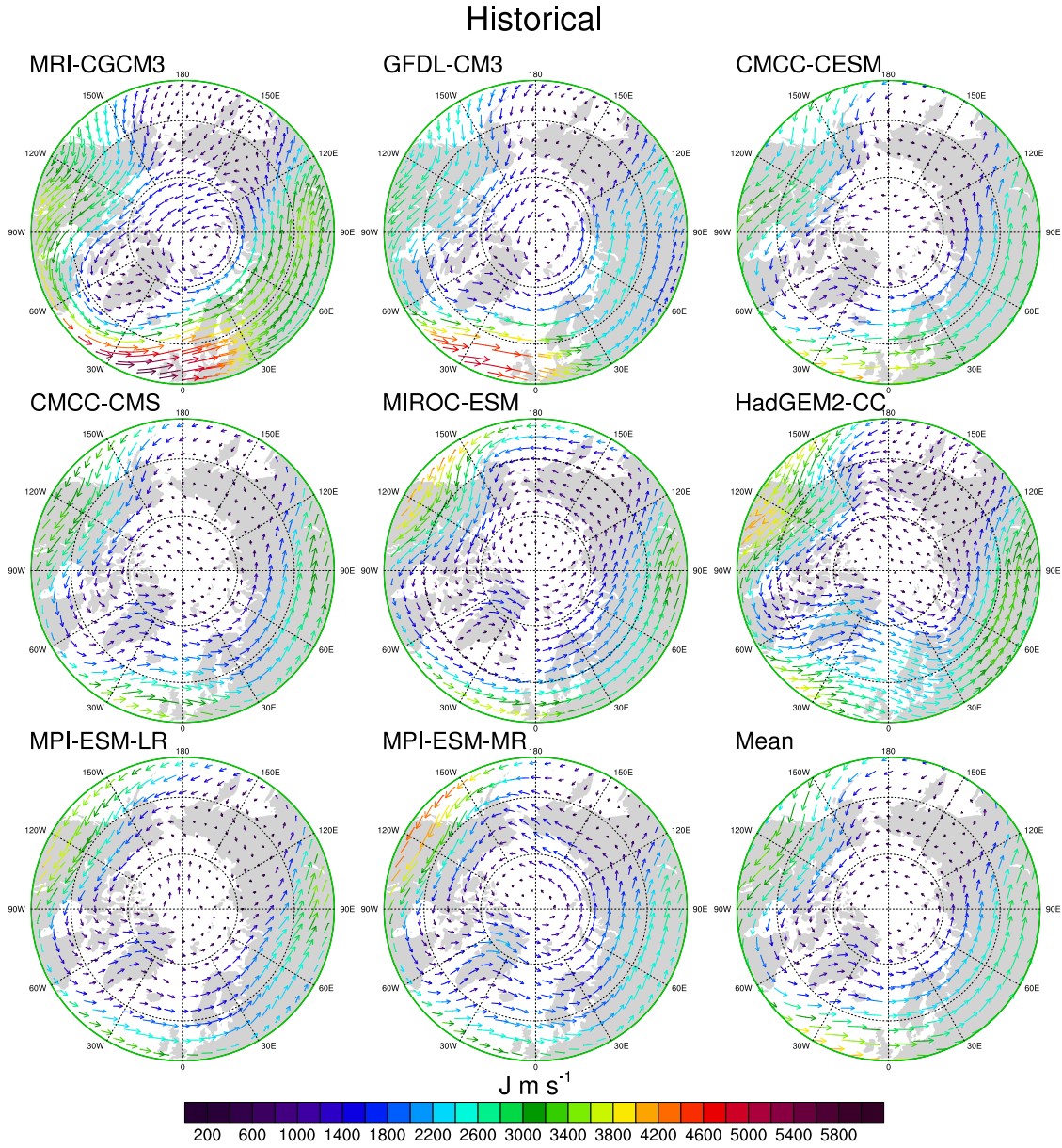

**Figure 2.** Same as in Figure 1, but for the Continental pathways.

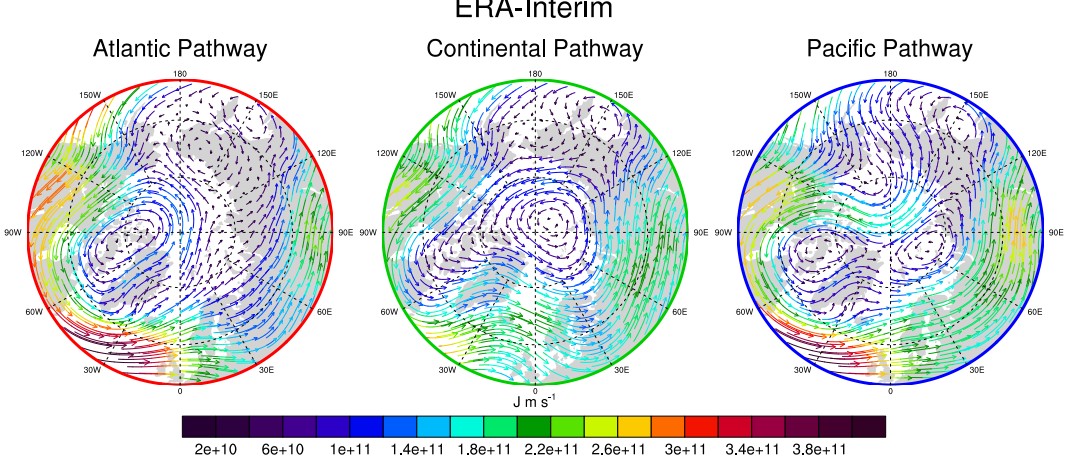

**Figure 3.** Same as in Figure 1, but for the Pacific pathway.

**Figure 4.** ERA Interim pathways modified from Figure 2 of Mewes and Jacobi [20].

These differences have been shown to be connected with very distinct meteorological conditions [20,43] like surface temperature, cloud properties, and downward longwave radiation. Thus, these pathways are a viable tool for distinguishing and analyzing meteorological states.

### 3.2. RCP8.5 Patterns

In addition to the patterns and pathways derived from the historical time frame, we also calculated the SOMs from the RCP8.5 data (Figures 5–7). The individual patterns emerging from each SOM can be seen in the Supplementary Figures S9–S16. We decided to not show difference plots of the vectors, as they do not improve the presentation. Comparing these to the historical patterns, there are no structural differences, especially concerning the general directions of the temperature fluxes. The largest differences in the general directions or structures of the fluxes are visible in the HadGEM2-CC results for the Pacific pathway (compare Figures 7 and 3). There, a weak transport over east Siberia turns into a cyclonic structure centered over the East Siberian Sea.

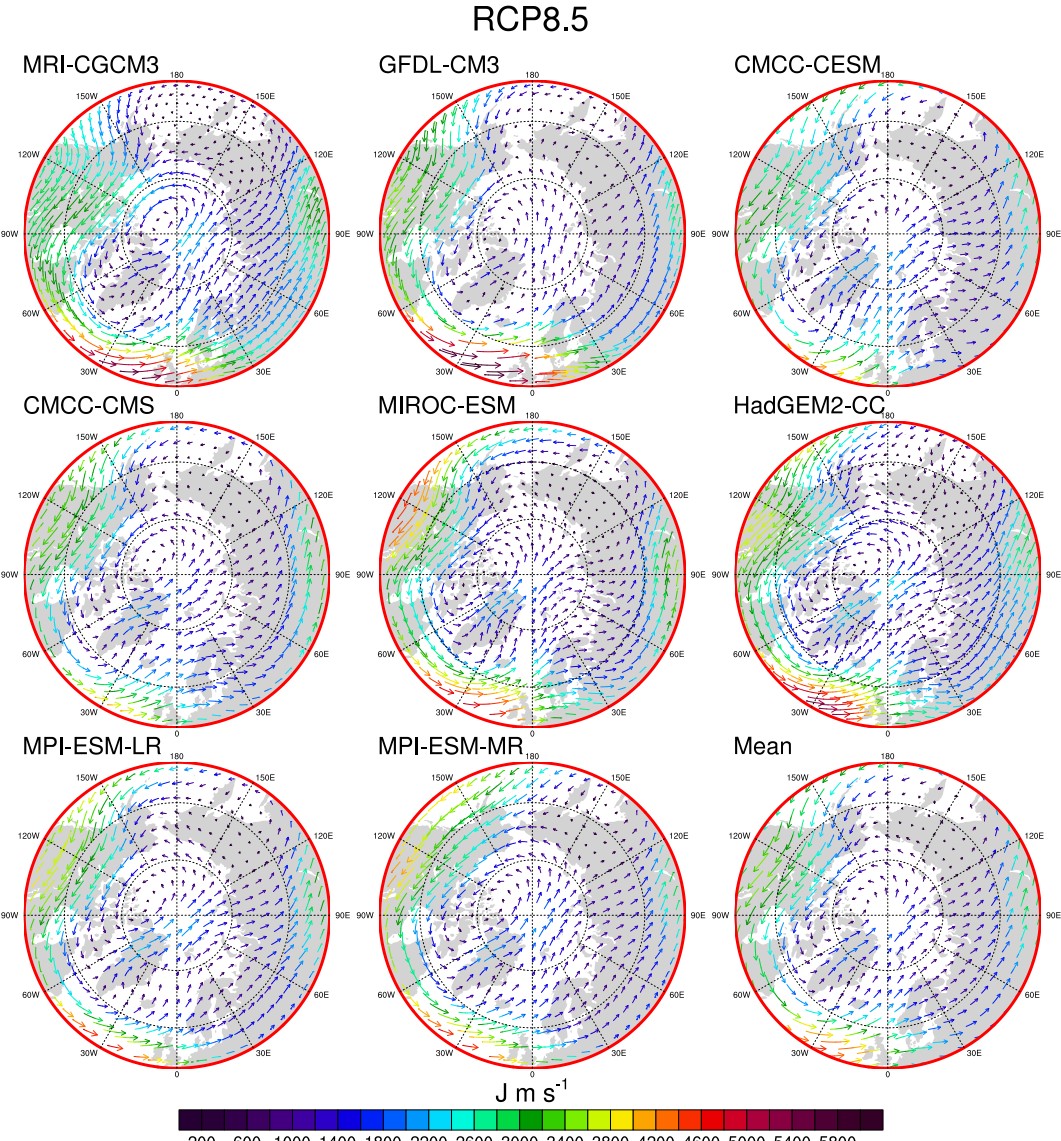

**Figure 5.** The Atlantic pathway for each of the analyzed models from Table 1, derived based on the CMIP5 RCP8.5 data for the winters from 2050–2099. Colors and arrows show the amplitude and direction of the horizontal temperature flux at 500 hPa. The multi-model mean of all pathways is shown in the bottom-right map.

The main differences of the RCP8.5 time frame results in comparison with those of the historical time frame occur in the general amplitudes of the transports. In Section 4, more objective measures will also be discussed. The RCP8.5 pathways produce smaller meridional fluxes into the Arctic compared to the historical pathway, which is especially the case for the Atlantic pathway (compare Figures 1 and 5). This indicates that the overall meridional flux amplitudes might decrease in the RCP8.5 scenario. Again, the Pacific pathway represented in HadGEM2-CC is an exception to that and shows stronger amplitudes (Figure 7). However, when focusing at the mid latitudes (50° N to 60° N), there is an increase in amplitude for the Continental pathway (compare Figures 6 and 2). This indicates stronger zonal circulation at mid latitudes in cases where the Continental pathway dominates.

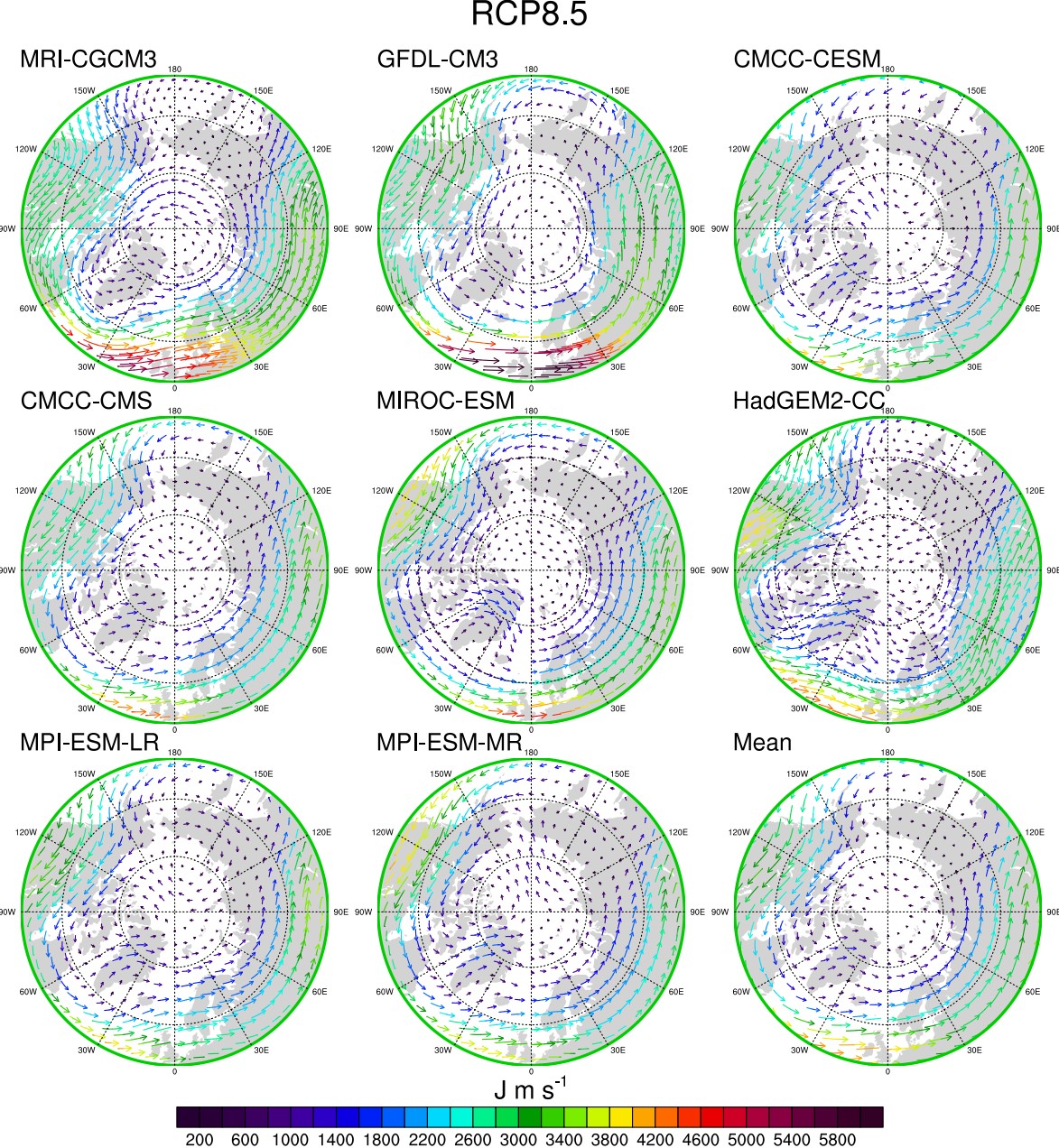

**Figure 6.** Same as in Figure 5, but for the Continental pathways.

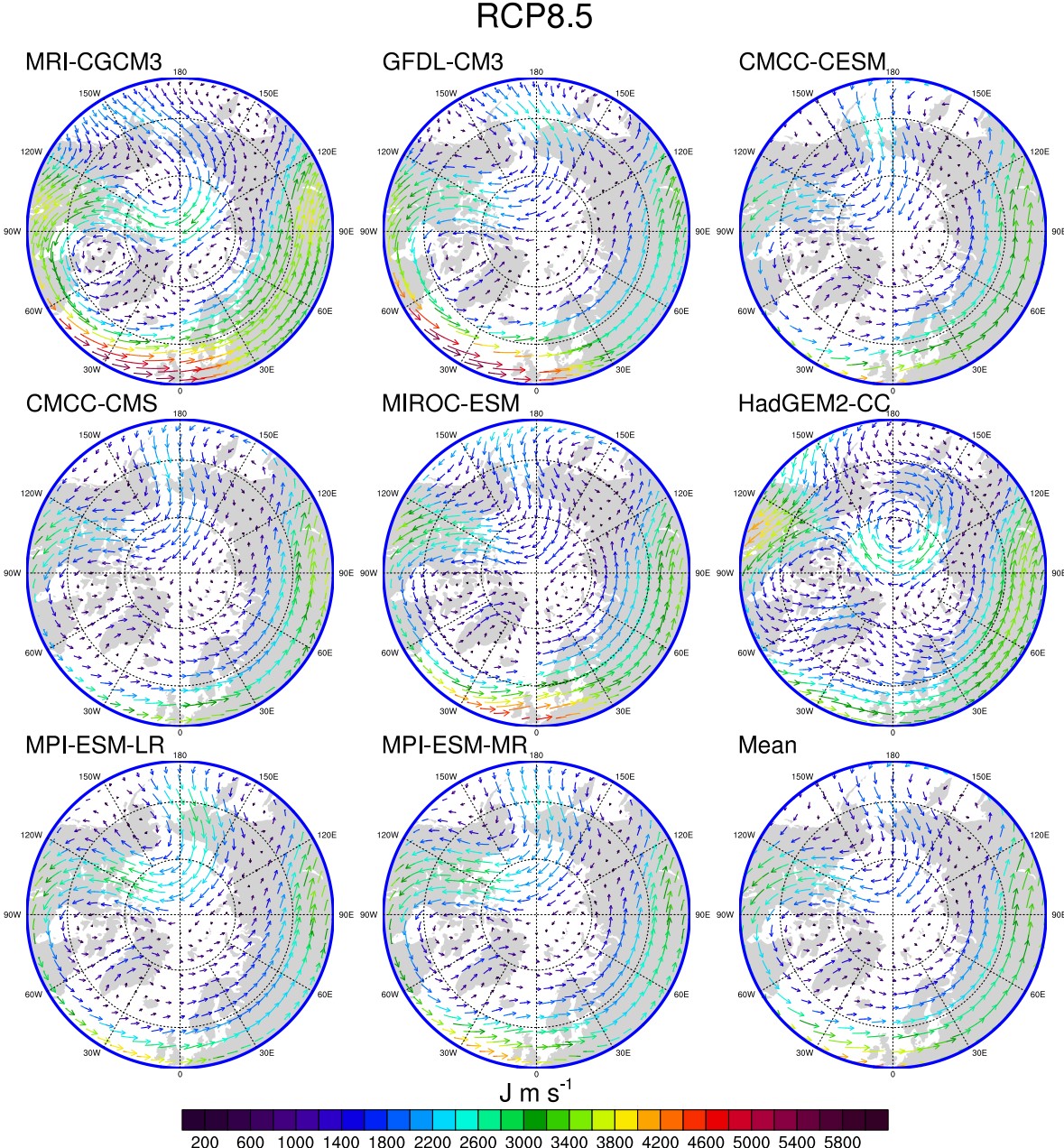

**Figure 7.** Same as in Figure 5, but for the Pacific pathways.

### 3.3. Mean Pathway Occurrence Frequencies

The average occurrence frequencies for the pathways as derived by each of the models for the historical experiment are shown in the first block of Table 2, together with the means of all models in the last column. The mean values of the occurrence frequencies show that in the historical experiment, the Continental pathway is most common, with about 37%. The miscellaneous pathways (superpositions) occur least often, with about 18%. The Atlantic and Pacific pathways show similar occurrence frequencies, which amount to 22% and 24%, respectively.

Comparing the specific models with the mean values shows larger differences. For the Atlantic pathway, the MIROC-ESM shows only 9% occurrence frequency, but much a stronger influence of the Continental (42%) and the miscellaneous pathways (25%). Like MIROC-ESM, the CMCC-CMS produces mostly fluxes connected with the Continental pathway. In CMCC-CESM and HAdGEM2-CC, the most frequent pathways are the miscellaneous ones, so superpositions of the three defined

pathways are most frequent (32% and 34%, respectively). GFDL-CM3 is the only model which predicts the most frequent transports through the Atlantic pathway.

**Table 2.** Relative occurrence frequencies of each pathway for each model of the historical time frame, and the relative occurrence frequency differences between Representative Concentration Pathway 8.5 (RCP8.5) based on historical pathways and on RCP8.5 pathways. All numbers are shown in %.

| Pathway | MRI-CGCM3 | GFDL-CM3 | CMCC-CESM | CMCC-CMS | MIROC-ESM | HadGEM2-CC | MPI-ESM-LR | MPI-ESM-MR | Mean |
|---|---|---|---|---|---|---|---|---|---|
| historical | | | | | | | | | |
| Atlantic | 23.27 | 30.76 | 26.32 | 17.00 | 8.73 | 24.95 | 24.68 | 17.88 | 21.70 |
| Continental | 43.25 | 25.99 | 25.62 | 43.09 | 41.97 | 33.16 | 44.84 | 38.91 | 37.11 |
| Pacific | 24.98 | 26.86 | 15.79 | 31.68 | 24.29 | 7.63 | 21.86 | 35.09 | 23.52 |
| Misc. | 8.49 | 16.38 | 32.27 | 8.23 | 25.01 | 34.27 | 8.62 | 8.13 | 17.68 |
| RCP8.5 based on historical minus historical | | | | | | | | | |
| Atlantic | −2.86 | −5.40 | −1.36 | −1.11 | −1.37 | 0.04 | −0.32 | −0.60 | −1.62 |
| Continental | −4.62 | 0.54 | 0.82 | 6.17 | 3.94 | 1.46 | 1.49 | −3.25 | 0.82 |
| Pacific | 5.94 | 3.70 | 0.79 | −3.60 | −6.43 | 0.50 | −0.36 | 0.68 | 0.15 |
| Misc. | 1.54 | 1.15 | −0.25 | −1.47 | 3.85 | −2.00 | −0.80 | 3.17 | 0.65 |
| RCP8.5 minus historical | | | | | | | | | |
| Atlantic | 7.39 | −4.57 | 7.31 | −0.85 | 16.54 | 1.09 | 1.27 | 8.07 | 4.53 |
| Continental | −22.88 | 13.33 | 2.12 | 8.10 | −14.61 | −8.43 | −20.46 | −14.53 | −7.17 |
| Pacific | 7.61 | −1.46 | 16.59 | −15.47 | −9.15 | 17.84 | 10.09 | −3.14 | 2.86 |
| Misc. | 7.89 | −7.30 | −26.02 | 8.22 | 7.22 | −10.50 | 9.11 | 9.61 | −0.22 |

Furthermore, the middle block of Table 2 shows the differences between the RCP8.5 occurrence frequencies based on the historical SOMs minus the historical SOM occurrence frequencies. This is used to derive the change of temperature flux, based on the assumption that these transport patterns do not change. The differences in occurrence frequencies are small (reaching from 0.82 to −1.62%) between the historical frequencies and RCP8.5 frequencies based on the historical SOM. As already shown in Section 3.2, the amplitudes of the horizontal temperature fluxes differ between the historical and RCP8.5 data.

To clarify, in the following paragraphs, we show the results based on the RCP8.5 SOMs and their derived pathways. This is helpful and necessary for taking into account that the general horizontal fluxes within the RCP8.5 projections have changed compared to the historical experiment. These differences are shown in the last block of Table 2. By comparing the historical pathway occurrence frequencies with the RCP8.5 pathway occurrence frequencies, the changes become stronger. The mean values of the occurrence frequencies show that the Continental pathway is still the most frequent, but its frequency is reduced by about 7%. The Atlantic pathway occurs more often by about 5%, and the Pacific pathway by about 3%. The miscellaneous pathways' occurrence frequencies for the RCP8.5 time frame are very similar to those of the historical time frame (only 0.22% less).

Comparing the different models, there is a large discrepancy compared to the mean occurrence frequencies. The biggest differences occur for CMCC-CESM, which reduces the miscellaneous pathways by about 26% and increases the occurrence frequencies of the Pacific pathway by about 17%. In addition, HadGEM2-CC shows a decrease in occurrence frequencies for the miscellaneous pathways (−11%) and the Continental pathway (−8%), as well as an increase of the Pacific pathway by about 18%. However, there are also models that do not show a decrease in the Continental and miscellaneous pathway occurrences. For example, CMCC-CMS shows a decrease in the Pacific pathway (−15%) and an increase in the Continental (8%) and miscellaneous (8%) pathways. The GFDL-CM3 shows the strongest increase in the Continental pathway with 13%.

Table 3 shows the V-measure to compare the differences between the RCP8.5 data mapped onto historical SOMs and onto RCP8.5 SOMs. This comparison shows if the time series of the RCP8.5 data is comparable between the approaches of neglecting circulation changes or not by comparing the two different mappings/projections of/on the RCP8.5 and historical SOMs. A V-measure of 1 would mean that every day of the RCP8.5 time frame would have been mapped onto the same pattern in both cases. All models show similar V-measure values between 0.37 and 0.48. The completeness scores and the homogeneity scores are within these ranges too, and do not provide clear differences in the classification. These values suggest that the general changes in clustering are similar across all models.

Consequently, all models reveal changes in circulation of similar strengths between the historical clustering and the RCP8.5 clustering. Even though the change across all models is not uniform, this suggests that the differences in historical and RCP8.5 boundary conditions result in similarly strong changes in the transports shown here.

**Table 3.** V-measures [41] derived from two different mappings of RCP8.5 data mapped on RCP8.5 SOMs and on historical SOMs. A V-measure equal to 1 would mean that both approaches group the RCP8.5 identically.

| Model | V-measure | Homogeneity | Completeness |
|---|---|---|---|
| MRI-CGCM3 | 0.43 | 0.43 | 0.42 |
| GFDL-CM3 | 0.40 | 0.41 | 0.40 |
| CMCC-CESM | 0.40 | 0.40 | 0.40 |
| CMCC-CMS | 0.39 | 0.39 | 0.39 |
| MIROC-ESM | 0.37 | 0.37 | 0.36 |
| HadGEM2-CC | 0.38 | 0.38 | 0.38 |
| MPI-ESM-LR | 0.48 | 0.48 | 0.48 |
| MPI-ESM-MR | 0.42 | 0.42 | 0.41 |

*3.4. Pathway Occurrence Frequency Trends*

Figures 8–10 show the trends in occurrence frequency for the historical time frame. The trends are shown for each model and the three major pathways. Additionally, we calculated a multi-model mean trend. Note that the trends are not independent of each other. For each year, the total sum of the occurrence frequencies among the Atlantic, Continental, Pacific, and miscellaneous pathways has to add up to one hundred percent. This means that if one pattern shows a trend, this trend has to be counteracted by the other pathways' trends.

Models like GFDL-CM3 and CMCC-CMS do show positive trends for the Atlantic pathway (Figure 8), but do not agree on the other pathways' trends in the historical time frame. Other models produce negative trends for the Atlantic pathway (Figure 8: CMCC-CESM, HadGEM2-CC). The largest positive trend for the Atlantic pathway is estimated by the MRI-CGCM3 with about 2.9% per decade, while MPI-ESM-MR and HadGEM2-CC show a maximum negative trend of about −1% per decade. The trends of the Continental pathway also differ between the models (Figure 9). MIROC-ESM shows the largest positive trend with about 1.5% per decade, while the most negative trend for the Continental pathway is shown by the CMCC-CESM with about −1.4% per decade. The Pacific pathway has either no (Figure 10: CMCC-CMS, MPI-ESM-MR) or a negative (Figure 10: MIROC-ESM, MRI-CGMC3) trend. The largest negative trend is shown by the MRI-CGCM3 with about −2.4% per decade. The trends shown for the multi-model mean are very small compared to the single models. The Atlantic pathway mean trend is about 0.4% per decade, the Continental pathway mean trend is about 0.3% per decade, and the Pacific pathway mean trend is about −0.8% per decade.

The trends have also been calculated for the RCP8.5 time frame based on the RCP8.5 SOMs (Figures 11–13). Largest positive trends of the Atlantic are shown by CMCC-CMS with about 2.2% per decade, and largest negative trends for this pathway are calculated by the MPI-ESM-MR with about −0.8% per decade (Figure 11). The Continental pathway trends differ strongly between the models. The largest positive trend with about 1.8% per decade is shown by MIROC-ESM, while the largest negative trend with −2% per decade is calculated by CMCC-CMS (Figure 12). MRI-CGCM3 shows the strongest positive trend for the Pacific pathway with about 0.1% per decade, while the largest negative trend is shown by MIROC-ESM with −1.9% per decade (Figure 13).

The multi-model mean trends are smaller compared to the individual models. The mean trend of the Atlantic pathway is about 0.3% per decade, the Continental pathway's mean trend is about −0.1% per decade, and the Pacific pathway's trend is about −0.7% per decade.

# Historical - Atlantic Pathway

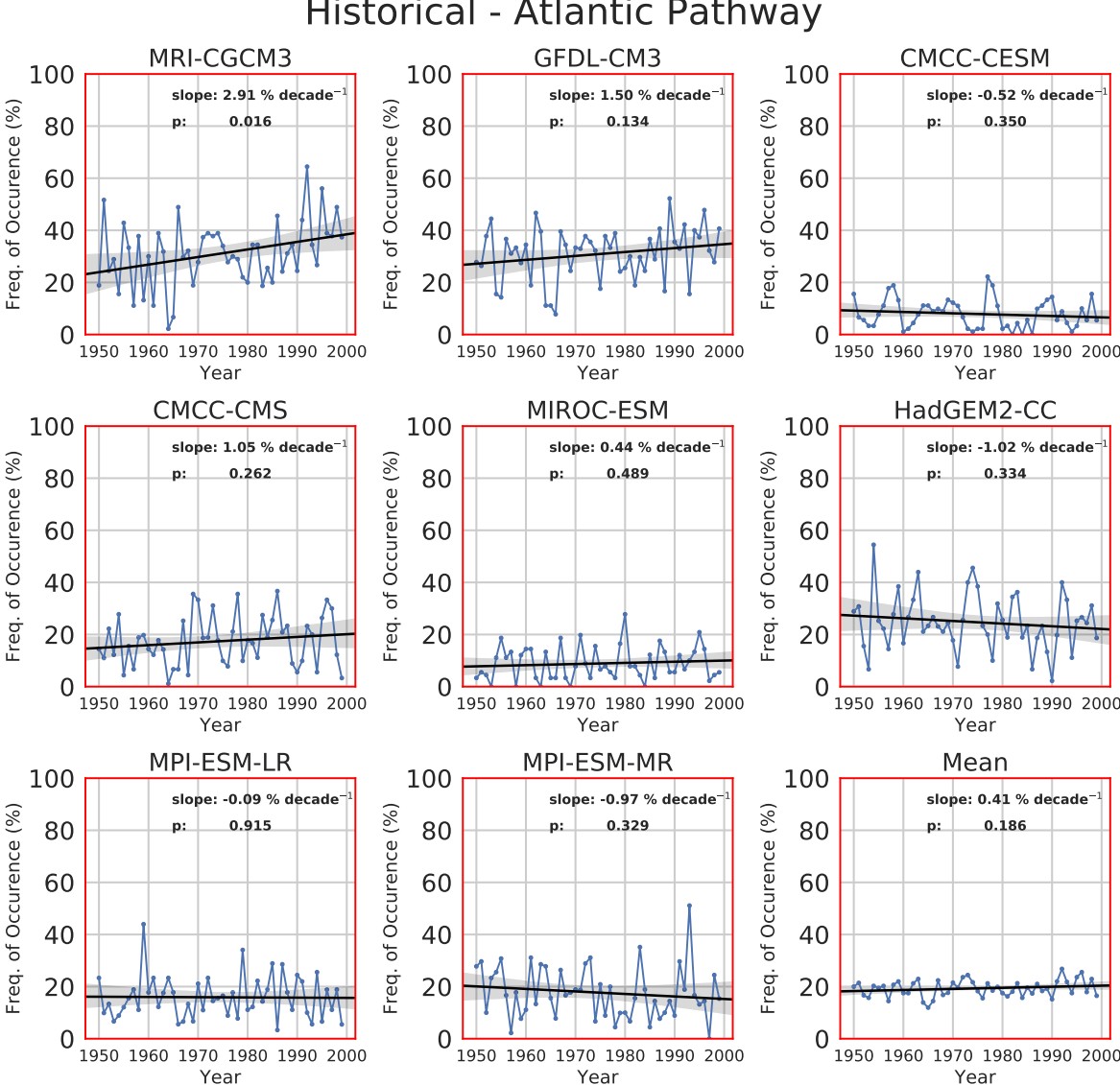

**Figure 8.** Relative occurrence frequencies for the Atlantic pathway for each model and a multi-model mean derived for each winter (blue line) of the historical time frame (1950 to 1999). The linear trend is shown as black line, and greyshading shows the 95% confidence intervals for the trends derived by bootstrap resampling. *p* values are shown according to a two-sided *t*-test.

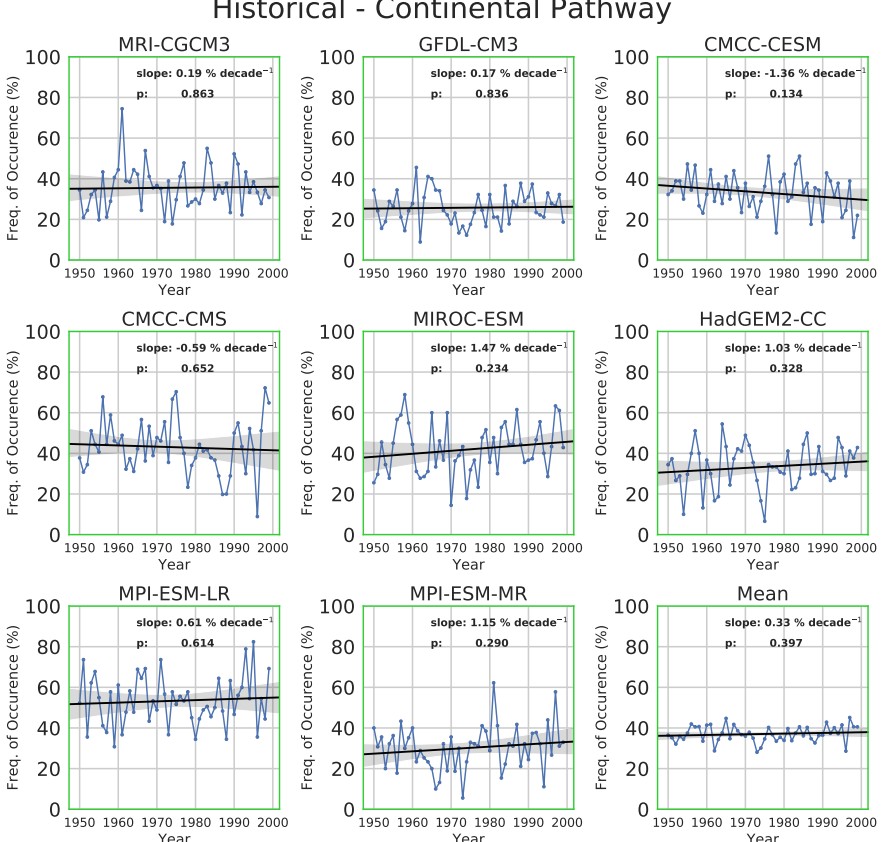

**Figure 9.** Same as in Figure 8, but for the Continental pathway.

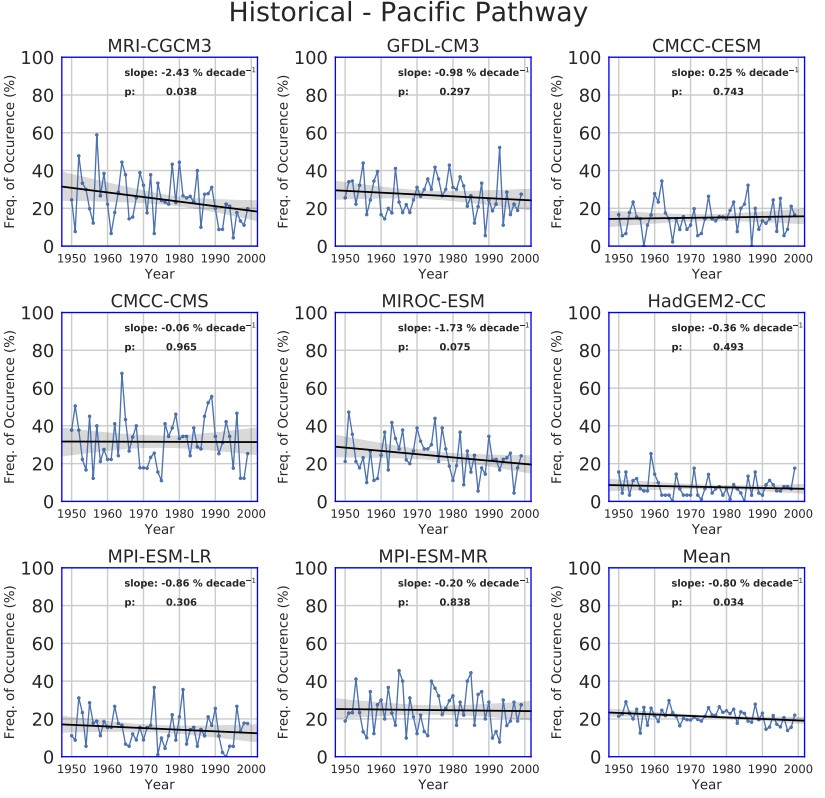

**Figure 10.** Same as in Figure 8, but for the Pacific pathway.

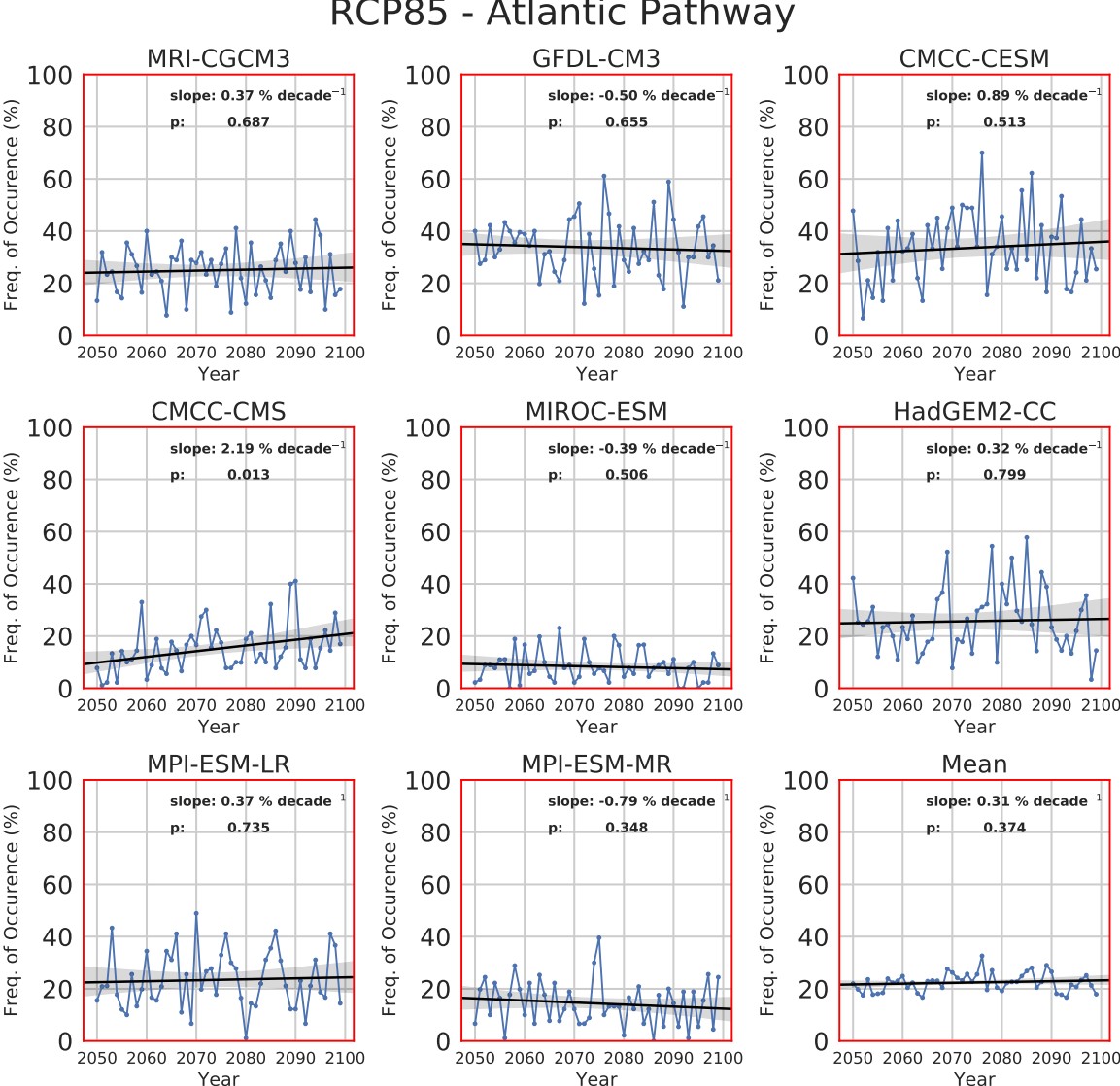

**Figure 11.** Atlantic pathway's relative occurrence frequencies for each model and a multi-model mean derived for each winter (blue line) of the RCP8.5 time frame (2050 to 2099). The linear trend is shown as black line, and greyshading shows the 95% confidence intervals for the trends derived by bootstrap resampling. *p* values are shown according to a two-sided *t*-test.

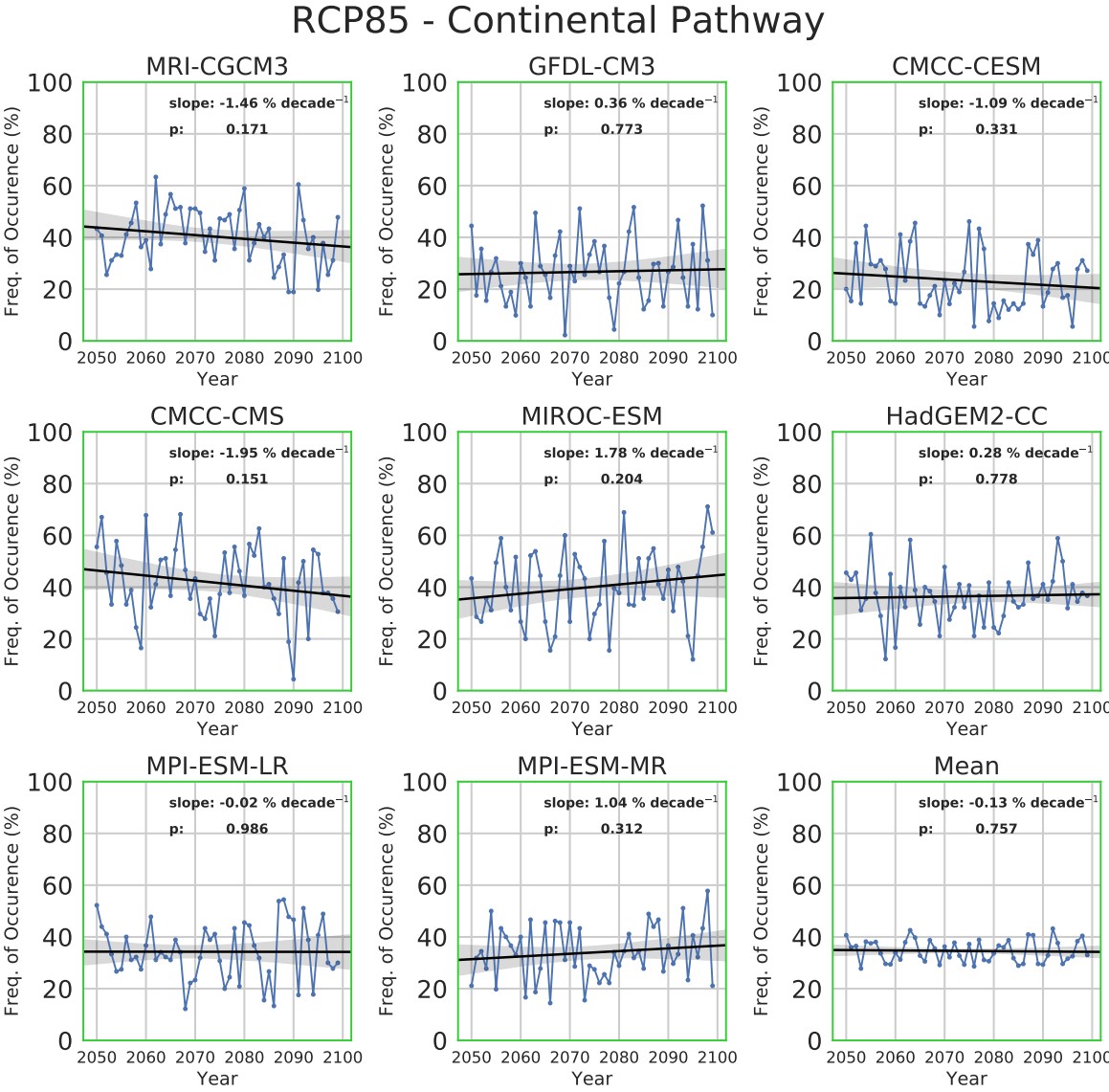

**Figure 12.** Same as in Figure 11, but for the Continental pathway.

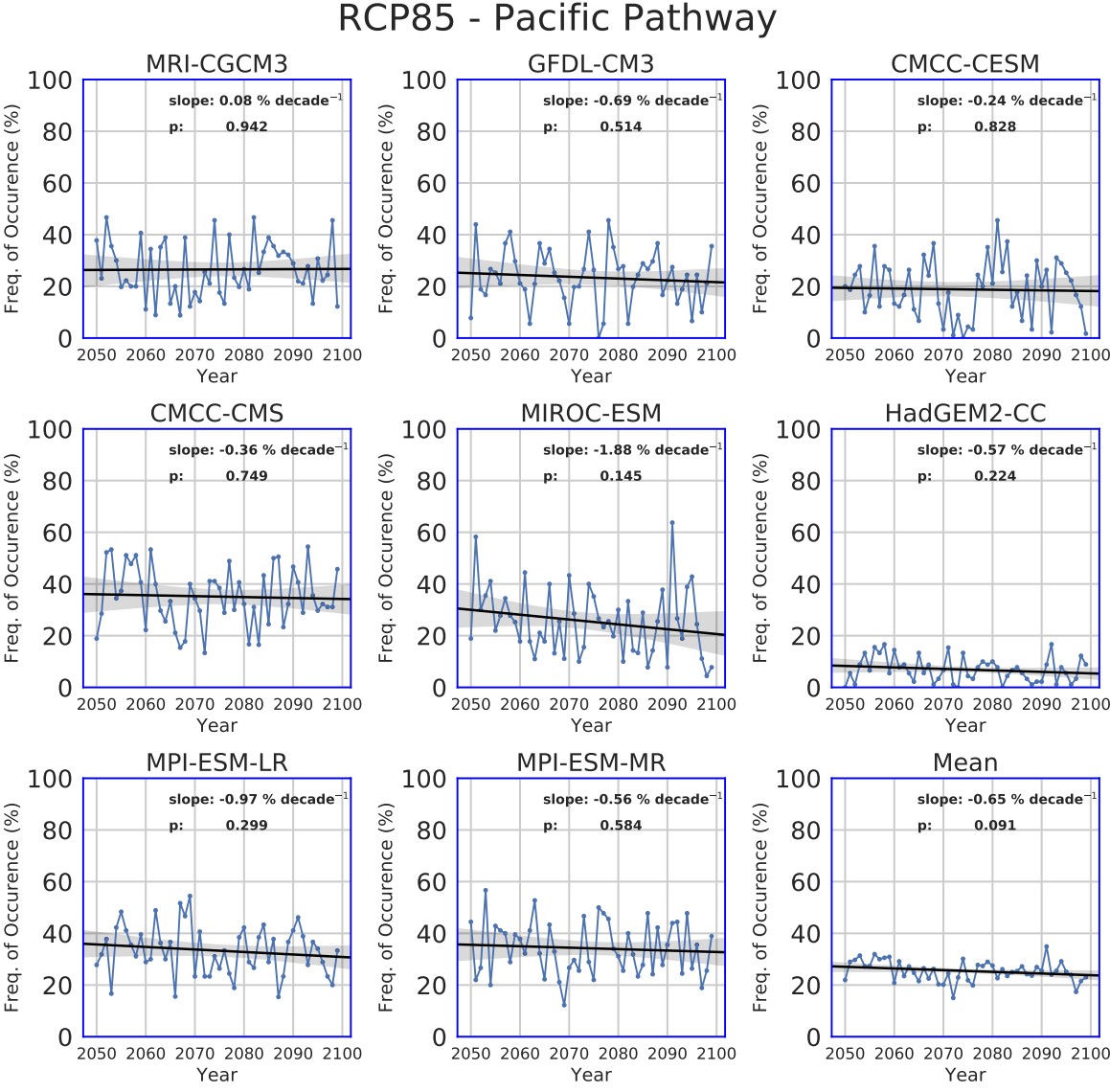

**Figure 13.** Same as in Figure 11, but for the Pacific pathway.

## 4. Discussion and Conclusions

The general structures of temperature flux pathways at 500 hPa are comparable to the general structure found in the ERA Interim reanalysis by Mewes and Jacobi [20]. They used the horizontal vertically integrated moist static energy flux from the ERA Interim reanalysis for the winters 1979 to 2016. The similarity between their results and our maps suggests that the temperature flux at 500 hPa might be a good proxy for qualitatively characterizing different flux/transport pathways while using the SOM method. This leads to the conclusion that the general shape of the horizontal heat flux is not very different between these models and the reanalysis. This further suggests that the general underlying dynamics shown by reanalyses are captured in the models. In addition to the three pathways found by Mewes and Jacobi [20], we found superpositions of these pathways that are present in each model. The general relative occurrence frequencies differ when comparing the historical time frame with the analysis of Mewes and Jacobi [20]. Furthermore, the strict grouping into only three major pathways of Mewes and Jacobi [20] was done due to the lack of knowledge about multiple patterns that favor different kinds of superpositions, which have now been shown in the analysis of the CMIP5 data. This is why we introduced the miscellaneous patterns.

The CMIP5 historical results show that the most frequent pathway is the Continental pathway, while for the ERA Interim, it was shown that the Pacific pathway occurs most often (their fig. 2 [20]). This representation of the strong zonal flow might be induced by the lack of a well-resolved stratosphere. The stratosphere plays a key role in the dynamics in the Arctic and the Northern Hemisphere [32,33], and its insufficient representation might produce errors in the projection of the dynamics. Furthermore, Kim and Kim [44] showed in the ERA Interim winter data from 1979–2017 that the vertical flux also explains a considerable fraction of the increase of the mean in temperature. This shows that the vertical exchange of latent and kinetic energy, as well as the exchange between the lower and the middle atmospheres, is also an important factor to consider when describing the dynamics of the troposphere. However, the large-scale horizontal advection is the major source of variability in the temperature of the atmospheric column.

For the Siberian region, studies have shown that the amount of precipitation increases in the RCP8.5 scenario [45]. The flux features of the Pacific pathway over Siberia might bring moist air into this region and thus increase precipitation.

By comparing the CMIP5 historical simulations for the years 1976–2005 and the respective ERA Interim reanalysis results (1979–2008), Zappa et al. [46] showed that storm tracks in CMIP5 simulations tend to be either too zonal or displaced southward compared to the ERA Interim reanalysis. This coincides with our results when comparing the occurrence frequencies of the respective North Atlantic pathway and the Continental pathway. We found that CMIP5 models favor zonal transports as opposed to ERA Interim, where meridional transports through the North Pacific and North Atlantic are more frequent. However, different models also show different kinds of superpositions in their respective SOMs.

Furthermore, the difference between the reanalysis and the CMIP5 results shows that the latter does not necessarily correctly describe the current development of the Earth's system. Vihma et al. [47] showed that the large-scale circulation has a strong impact on the European winter temperature during periods of amplified Arctic warming. Insufficient description of the former may result in errors of the representation of the atmospheric state. Furthermore, the CMIP5 results have been shown to insufficiently describe the downward propagation of stratospheric anomalies into the troposphere [48]. This anomaly propagation is very important for the Northern Hemisphere's winter season circulation, such as the storm tracks and the strength of the mid-tropospheric flow [49]. It can be assumed that the CMIP6 [50] model results might fit better to reanalyses for the historical time frame.

We compared two different approaches concerning the RCP8.5 time frame. First, we projected the historical SOMs onto the RCP8.5 time frame and thus assumed that the patterns would not change between the historical and RCP8.5 time frames. Second, we created the SOMs and thus the pathways directly from RCP8.5, and, in this way, were able to find the patterns and pathways that are present within the RCP8.5 simulations. As described above, the two approaches show differences in the amplitudes and occurrence frequencies. This might indicate that historical and RCP8.5 data show distinct differences in the transports. Another possibility is that the historical SOMs were overfitted to the historical time frame and were not suitable to be used with the RCP8.5. Concerning the use of the historical SOMs as a base for the RCP8.5 time frame, it might be possible that the SOM of the horizontal temperature flux might not be a good proxy for the projection of the total horizontal energy transport in CMIP5 experiments. However, our analysis was not focused on the application of the SOM as a tool for prediction, but rather for clustering atmospheric states of two different CMIP5 experiments.

The CMIP5 historical mean pathway occurrence frequencies show that the most frequent pathway is the Continental pathway, while in Mewes and Jacobi [20], the Continental pathway is the least frequent. The mean occurrence frequency of the Continental pathway decreased in the RCP8.5 experiment in comparison to the historical ones, but still remains the most frequent pathway. In context with the general decrease in meridional flux amplitudes of the RCP8.5 experiment, this decrease in frequency can be expected in the Continental pathway. As the general amplitude decreases, the occurrence of respective meridional transport events increases. This is necessary, as the total

amount of energy transported must be conserved. Thus, more frequent transport events result in lower flux amplitudes. As already stated, the CMIP5 model results of the mean occurrence frequencies do not match the general representation of the ERA Interim occurrence frequencies. This suggests that the CMIP5 models insufficiently reproduce the atmospheric state compared to long-time ERA Interim data. However, the general differences between historical mean occurrence frequencies and RCP8.5 show that, in general, the projections of the models show an increase of more meridional fluxes (Atlantic and Pacific pathway) while reducing the amount of more zonal structures (Continental pathway). A physical explanation would be the reduced meridional temperature gradient between higher and middle latitudes, which results in a weaker jet stream. This reduced strength in the jet stream would lead to more meandering and thus more meridional fluxes, which explains the increase of occurrence frequencies of the Atlantic and Pacific pathways.

The phenomenon that models favor zonal flow and less perturbation of the transports/fluxes has also been shown by Hanna et al. [51]. They showed that the variability of the North Atlantic Oscillation (NAO) in the CMIP5 RCP8.5 data does not change with time, but that there is a trend towards a more positive NAO. This observed positive NAO trend by Hanna et al. [51] does not fit with our results, as a more perturbed flow does not fit well with the increased NAO index in the RCP8.5 model results.

The discrepancy between the reanalysis models and the CMIP5 results was also shown in Belleflamme et al. [52]. They showed that for the summers from 1961 to 1990, most of the CMIP5 models could not represent the persistence and frequency of the circulation types found in the reanalyses. This can be also confirmed in our studies. In general, the circulation structures can be represented by the models in comparison with the reanalyses [20]. However, the general occurrence frequencies of those typical structures differ substantially between the models and the reanalyses.

By analyzing CMIP5 RCP4.5 results from 1979 to 2013, García-Serrano et al. [53] showed that sea ice reduction over the eastern Arctic is followed by negative NAO-like patterns. This conforms with our results with respect to the larger meridional transports in the RCP8.5 data compared to the historical data, as the sea-ice cover is generally lower in RCP8.5 (e.g., [54], their fig. 1) compared to RCP4.5 and the historical analyses (e.g., [55], their fig. 2).

Another model study using the Whole Atmosphere Community Climate Model showed that, by the loss of sea ice from August to November, the polar vortex shows a weakening in the following January [56]. The impacts of this weaker vortex generally resemble a negative Northern Annular Mode (NAM) phase during winter, and thus weaker zonal flow, which is in general agreement with our findings when comparing historical and RCP8.5 results.

Yu et al. [57] used the SOM method for clustering the anomalous daily sea-ice concentration during the melt season from June to August for the period 1979–2016 using sea-ice data from the U.S. National Snow and Ice Data Center and ERA Interim reanalysis. They found that a positive sea-ice anomaly is, among other factors, connected with a more zonal anomaly of the 850 hPa wind vector. Negative sea-ice anomalies have been connected with more meridional anomalies of the 850 hPa wind vector. This also shows that negative sea-ice anomalies occur in a more meandering jet stream, which might be connected with a negative NAO.

The mean occurrence frequency trends for the historical and the RCP8.5 data both show an increase for the Atlantic pathway. The RCP8.5 results show a small negative trend for Continental pathway, while the historical mean trend is negative. The Pacific pathway shows a negative mean trend in both experiments. Note that in both experiments, the multi-model mean Atlantic pathway occurrence frequency starts at similar values in the historical and RCP8.5 cases. This similarity in the starting occurrence frequencies might result from a similar phase of the Atlantic multidecadal oscillation [58] for the given historical and RCP8.5 time frames.

Other factors that have not been addressed within this paper are the accumulated changes of the horizontal fluxes or the persistence of respective patterns and the interpretation of their influences on the atmospheric column (e.g., [43]). As such an analysis was beyond the scope of this paper, a future

analysis of whether more or less persistent patterns may quantitatively contribute to temperature changes is of interest for analyzing their role in Arctic climate change.

We may conclude that the general structures of horizontal fluxes are similar between the reanalysis and the CMIP5 results. Furthermore, we can state that all models show similar amounts of differences between the historical and the RCP8.5 cluster mappings. This leads to the suggestion that the models' circulations show changes in similar strength but not in a uniform way. These non-uniform changes between the models might be caused by the lack of representation of the stratosphere and thus the misrepresentation of internal variability. For future analysis, it will be interesting if the connected temperature anomalies of these pathways change within a changing climate as well.

**Supplementary Materials:** The following are available online at http://www.mdpi.com/2073-4433/11/3/251/s1.

**Author Contributions:** D.M. performed the data analysis and wrote the first draft of the manuscript. C.J. initiated the project and made contributions to the interpretation of the results and writing of the manuscript. All authors have read and agreed to the published version of the manuscript.

**Funding:** We gratefully acknowledge the funding by the Deutsche Forschungsgemeinschaft (DFG, German Research Foundation)—project number 268020496—TRR 172, within the Transregional Collaborative Research Center "ArctiC Amplification: Climate Relevant Atmospheric and SurfaCe Processes and Feedback Mechanisms $(AC)^3$" in the sub-project D01.

**Acknowledgments:** We acknowledge the World Climate Research Program's Working Group on Coupled Modeling, which is responsible for CMIP. We thank the climate modeling groups (listed in Table 1 of this paper) for producing and making their model output available. For CMIP, the U.S. Department of Energy's Program for Climate Model Diagnosis and Intercomparison provides coordinating support and led development of the software infrastructure in partnership with the Global Organization for Earth System Science Portals.

**Conflicts of Interest:** The authors declare that they have no conflict of interest. The funders had no role in the design of the study; in the collection, analyses, or interpretation of data; in the writing of the manuscript, or in the decision to publish the results.

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
