# Peer review of "Horizontal Temperature Fluxes in the Arctic in CMIP5 Model Results Analyzed with Self-Organizing Maps"

_atmosphere, doi:10.3390/atmos11030251_

Round 1
Reviewer 1 Report
Title: Horizontal temperature fluxes into the Arctic in CMIP5 model results analyzed with Self-Organizing Maps
Manuscript Number: atmosphere-713751
Type of manuscript: Article
Authors: Daniel Mewes and Christoph Jacobi
Recommendation: conditionally accept
General Comments:
This study investigated the pattern of the atmospheric horizontal temperature flux and the general occurrence frequencies of the patterns in CMIP5 historical and RCP8.5 using the self-organizing-map method. In addition, authors showed the trends of the occurrence frequencies obtained from the individual models and their model mean. I think that the method which extract intrinsic circulation patterns using the SOM and explanation for characteristics of climate models were described properly. However, I guess this manuscript need to be corrected for some parts in the contents. So, I would like to conditionally accept this paper under correction of comments below.
Major comments:
Through the comparison between the pattern analysis of observation and CMIP5 historical results, assessment of the capabilities of the used models seems to be necessary. Is it possible to trust the results without knowing the performance of the used model? Even though authors mentioned that your study was focused on clustering atmospheric states of two different CMIP5 experiments, if you don’t know the model performance, how reliable are the mean values of the frequency of occurrence of the models used in the study, especially in the future simulations such as CMIP5 RCP8.5?
I know that there are lots of model simulations participated in CMIP5. What are the criteria and reasons for selecting the CMIP5 model used in this study?
In section 4, authors mentioned that the structures of temperature flux from CMIP5 are similar to observed structures by Mewes and Jacobi (2019). Is the comparison between the general structure of temperature flux obtained from ERA interim reanalysis by Mewew and Jacobi (2019) and authors’ results justified? They used 1979-2016 for observation. Authors used from 1950 to 1999 for historical CMIP5 data. I think authors should examine and compare with observed data for the same periods.
Minor comments:
Explanation for Section 2.2 Self-Organizing Maps is too much long. I think it would be fine if this part is summarized and the description of the SOM itself is attached in the Appendix or Supplementary if possible.
L168-175: Authors mentioned that the overall meridional flux amplitudes might decrease in the RCP8.5 scenario and the Pacific pathway (HadGEM2-CC) shows stronger amplitudes. However, I think it is difficult to actually distinguish them. Difference Figures are needed to prove exactly.
Figure 1: Caption is needed to correct.
“the CMIP5 RCP8.5 data for the winters from 1950–1999”
=> “the CMIP5 historical data for the winters from 1950–1999”
Numbers in the top right denotes the occurrence frequency for each pathway
=> Where is numbers?
Figure 4: Caption is needed to correct or frequency is need to add in Figure.
Numbers in the top right denotes the occurrence frequency for each pathway
=> Where is numbers?
L204-205: “The miscellaneous pathways’ frequency of occurrence are the same compared to the
historical time frame.”
=> I don’t understand. Authors need to correct clearly.
L208-209: In this sentence, “increases the occurrence frequency of the Pacific pathway by a factor of 5 compared to the mean.”
=> What is “a factor of 5”?

Reviewer 2 Report
Summary:
The authors followed the study of Mewes and Jacobi [2019] by applying the SOM method on CMIP5 historical scenario and RCP8.5 scenario. The purpose of the study can help the reader to understand the performance of energy flux on CMIP5 models. However, this manuscript does not clearly describe the methodology, and the selection of different pathways results from manually visual judgment from SOM outputs, which causes the performances of RCP8.5 and historical are totally different in the same model. Therefore, several points should be clarified and improved before the manuscript can be published.
Major points:
Why do the authors only use 8 models? The authors have to include more models, especially some good performance models, such as CESM1 (CCSM4), IPSL, CAN-ESM, etc.
The authors add one more category from the previous study, which makes the manually visual selection more complicated and inconsistent.
Moreover, because the category of historical and RCP8.5 SOM are manually grouped, it causes the performance of each model is not consistent, and the comparison between historical and RCP8.5 is not “chicken to chicken” comparison. I would suggest that the authors just keep three categories and focuses on Atlantic and Pacific pathways. The most important is that the authors could set up thresholds for different pathways and make the selection more objective.
Lines 52-57, Lines 113-115, Table2, and Table 3:
What are the two different scenarios? (historical and RCP85?)
How do the authors map RCP8.5 SOM on Historical SOM?
What is the difference between “RCP8.5 based on historical” and “RCP8.5”?
What are “two different approaches”? (RCP8.5 based on historical and RCP8.5?)
The authors should clearly describe the detail of the methodology. Moreover, the V-measure method should be mentioned in the method section or supplementary.
Lines 36-37: Plagiarism. Please rewrite and provide more information about the energy transport and mass transport and their influences.
Minor points:
Line 77: How do the authors limit the F (v T)? Please state the detail of the methods.
Line 111: twelve?
Lines 150-152: CMCC-CMS and CMCC-CESM present similar patterns. How do the authors distinguish them?
Figure S3: Follow point 3, I would like to know how the authors catalog the patterns. Can the authors more specifically reply here to talk about 1) why 1.1 is not a continental pathway, 2) why 1.1 is not a Pacific pathway, and 3) why 1.3 is not an Atlantic pathway if 2.3 is an Atlantic pathway?
Lines 171-173: HadGEM2-CC only shows one Pacific pathway in Figure S6. How many Pacific pathway of HadGEM2-CC are shown in RCP8.5? Please also add the SOM of RCP8.5 of each model in the supplementary. Moreover, why Figure S6 1.1 is not a Pacific pathway? It is similar to Figure 1 – 2.4 in Mewes and Jacobi [2019].
Reviewer 3 Report
Please find my comments in the attached document.

Reviewer 4 Report
This study investigates changes in midlatitude large-scale circulation patterns induced by the RCP8.5 climate change scenario. The circulation patterns are identified using a SOM method and the study analyzes 8 models participating in CMIP5. To characterize the large-scale circulation the study focuses on the horizontal heat transport at 500 hPa. The results show that models disagree about the slope of the trend of simulated changes. Another finding of the study is that circulation patterns mostly change their intensity in response to the transient warming. This is an interesting study that can be published after addressing my concerns:
Additional analysis based on reanalysis and model comparisons need to be included. The conclusion and discussion section lacks the supporting analysis. The interpretation of results is limited to description of figures with very little attempt to give a physical interpretation.Specific comments:
Figure 1 caption: RCP8.5 data-> historical data. Numbers showing the frequency of occurrence for each pathway are missing.
Figure 1: There are large differences between the patterns simulated by the models. Replacing the model mean by a pattern based on reanalysis will give the reader an idea of what model is closer to observations without having to consult another publication.
Figure 4 caption: Numbers showing the frequency of occurrence for each pathway are missing.
L168-169: It is hard to assess how different these vector plots are. Pattern correlation between the components of the fluxes or amplitudes will provide an objective measure of the differences. The discussion and conclusion section has many comments on this comparison. They will fit better in here.
L254-255: What is the meaning of model mean when models display both positive and negative trends?
L268-271: In addition to stratosphere there could be many other factors. This paragraph needs to be either expended to include other factors or removed.
L272-276: Please provide some references.
L285: I’m not sure what is the message here. Please rephrase.
L347-348: The analysis presented here does not use ERA Interim data, therefore such statement cannot be presented as a conclusion of the study.
Minor comments:
L192: “based under the assumption” -> under the assumption or based on the assumption
L228: “shows” -> show
L343: “both experiment” -> both experiments
Reviewer 5 Report
The submitted article is a continuation of research by the authors concerning heat transport to the Arctic, as described in detail among others in: Heat transport pathways into the Arctic and their connections to surface air temperatures. Atmospheric Chemistry and Physics 2019, 19, 3927–3937. doi:10.5194/acp-19-3927-2019.
The article has a clear and transparent structure. In my opinion, the applied methods and obtained results contribute new content to the interpretation of processes occurring in the atmosphere.
Please consider the comments below.
The figures presenting trends are largely illegible. I suggest for figures 1 and 4, 2 and 5, and 3 and 6 to be next to each other. It will facilitate their comparison. Alternatively, figures for mean values in two different periods of the analysis could be highlighted. Please expand the discussion part. Example papers:Revisiting the linkages between the variability of atmospheric circulations and arctic melt-season sea ice cover at multiple time scales, Journal of Climate, 2019
Relative role of horizontal and vertical processes in the physical mechanism of wintertime Arctic amplification, Climate Dynamics, 2019
The list is obviously longer. I encourage the authors to do further research.
I am surprised by lack of reference to earlier work of the authors: „Analyzing Arctic surface temperatures with Self Organizing-Maps: Influence of the maps size”, 2018. Please explain this issue. The article refers to future changes over the Arctic. Due to this I expect the authors to also discuss effects of accumulated changes (see bases: WoS, Scopus, etc.). It is not the goal of the article, but in my opinion it is interesting to potential readers. Please distinguish the conclusions chapter more clearly. Why are figures 10‑12 in between literature?Author Response
Please see the attachment.

Round 2
Reviewer 1 Report
For my requests, I think authors answered properly well. In this paper, even though authors could not insert or add any figures due to the insufficient space, I think authors need to show the results to the reviewer (in the reply) to agree with your opinions, not only by mention of your opinion as a few sentences. Anyway, I would like to accept the manuscript in this form.
Reviewer 2 Report
The authors have replied my concern.
I have no more comments and suggestions.
Reviewer 4 Report
The authors have satisfactorily addressed my main concerns.